# Association between fall-related serious injury and activity during fall in an acute care hospital

Kosuke Kobayashi[1]*, Naohiro Kido[1], Shoji Wakabayashi[1], Kyoko Yamamoto[1], Jun Hihara[2‡], Masami Tamura[2‡], Tomoko Sakahara[2‡]

1 Department of Rehabilitation, Hiroshima City North Medical Center Asa Citizens Hospital, Asa-kita Ward, Hiroshima City, Hiroshima, Japan, 2 Total Quality Management Center, Hiroshima City North Medical Center Asa Citizens Hospital, Asa-kita Ward, Hiroshima City, Hiroshima, Japan

☯ These authors contributed equally to this work.
‡ JH, MT and TS also contributed equally to this work.
* kobayashikosuke1126@gmail.com

## Abstract

### Objectives

Few studies have evaluated the mechanism of serious injury in acute hospitalization. Thus, the association between fall-related serious injury and activity during falls in acute care hospital remains unclear. Herein, we investigated the relationship between serious injury caused by fall and activity at the time of the fall in an acute care hospital.

### Methods

This retrospective cohort study was conducted at Asa Citizens Hospital. All inpatients aged 65 years and older were eligible for the study, which was conducted from April 1, 2021, through March 31, 2022. The magnitude of the association between injury severity and activity during the fall was quantified using odds ratio.

### Results

Among the 318 patients with reported falls, 268 (84.3%) had no related injury, 40 (12.6%) experienced minor injury, 3 (0.9%) experienced moderate injury, 7 (2.2%) experienced major injury. Moderate or major injuries caused by a fall was associated with the activity during the fall (odds ratio: 5.20; confidence intervals: 1.43–18.9, p = 0.013).

### Conclusion

This study recognizes that falling during ambulation caused moderate or major injuries in an acute care hospital. Our study suggests that falls while ambulating in an acute care hospital were associated not only with fractures, but also with lacerations requiring sutures and brain injuries. Among the patients with moderate or major injuries, more falls occurred outside the patient's bedroom as compared with patients with minor or no injuries. Therefore, it is

**Data Availability Statement:** All relevant data are within the paper and its Supporting Information files.

**Funding:** The authors received no specific funding for this work.

**Competing interests:** The authors have declared that no competing interests exist.

important to prevent moderate or major injuries related to falls that occur while the patient is walking outside their bedroom in an acute care hospital.

## Introduction

Fall is one of the most common health threats encountered in the aged population [1]. Falls occurring in the hospital are quite common with rates ranging from 2.3 to 17.1 falls per 1,000 patient-days [2, 3]. It is reported that approximately 23% to 42% of inpatient falls result in injury, and 4% to 8% result in serious injury [2, 4, 5]. Falls are a major problem for both patients and hospitals, resulting in increased length of hospital stay, discharge to a long-term care facility, and increased health care costs [6–8]. In addition, falls may also contribute to mental distress and decreased quality of life [9, 10]. Therefore, reduction and prevention of inpatient falls is an important patient safety issue for hospitals.

The Joint Commission recommends several actions to help health care organizations prevent falls and fall-related injury [11]. To reduce injuries due to falls, predicting the incidence of falls and implementing fall prevention activities are necessary [12]. However, little is known about how to effectively reduce falls in the hospital [13]. Perform a multifactorial falls risk assessment in all hospitalized older adults >65 years of age is recommended for fall prevention [14]. The effect of multifactorial interventions on the risk of falling is uncertain [15]. In addition, few studies have been conducted to identify predictors of harmful falls in hospitals [2]. Therefore, the mechanism of serious injury in acute hospitalization is unknown.

In the study conducted by Mayo et al, decreased mobility was associated with serious injuries in a rehabilitation hospital [16]. Although the only serious injury researched by those authors was fracture, several fall-related serious injuries exist. (e.g., cerebral hemorrhage, suturing, splinting of muscle/joint strain, etc.) Thus, the association between fall-related serious injury and activity during the fall in an acute care hospital is not clear. The objective of this study was to investigate the relationship between serious injury caused by fall and activity at the time of the fall in an acute care hospital. If we understand which patients are likely to become injured during a fall, we can prevent and reduce injuries caused by falling.

## Materials and methods

This retrospective cohort study was conducted at Asa Citizens Hospital, a 435-bed acute care hospital in Hiroshima. All inpatients aged 65 years and older were eligible for the study from April 1, 2021, through March 31, 2022.

An inpatient fall was defined as "an incident in which a patient suddenly and involuntarily came to rest upon the ground or other surface" [17]. Falls involving outpatients, visitors, and employees were excluded. Falls during physical therapy sessions were excluded because these patients are often exposed to situations in which they are more likely to fall. The 420 falls occurred in 347 patients, 50 of whom fell twice or more. Second or more falls were excluded to reduce bias for patients' characteristics and because patients who fell multiple times tend to repeat the type and location of the fall on successive falls [18]. Thus, we analyzed only the first falls for these 347 patients.

Fall-related data were collected from electronic medical records and fall reports. General characteristics included gender, age, and length of stay. Fall-related characteristics consisted of the following data; whether the fall was assisted, the location of the fall, shift when the fall occurred (day shift defined as 7 a.m. to 7 p.m.; evening/night shift defined as 7 p.m. to 7 a.m.),

fall occurring in the past 6 months, activity during the fall, and the type of severity of the injury. Through interviewing the patients, family or witness of the fall, identifying the activity during fall. Those whose activity during the fall was "unknown" were excluded. We excluded 29 falls because they were "unknown" and finally we used 318 falls for analyze.

Injury severity was classified as follows [19]. "None" indicated that the fall caused no significant discomfort, had no effect on clinical course, and did not result in an increased length of stay. "Minor" indicated that the fall resulted in the application of a dressing, ice cleaning of the wound, limb elevation, topical medication, bruise, or abrasion. "Moderate" indicated that the fall resulted in suturing, application of Steri-Stripes/skin glue, and splinting of muscle/joint strain. "Major" indicated that the fall resulted in surgery, casting or traction; required consultation for neurologic (basilar skull fracture, small subdural hematoma) or internal injury (rib fracture, liver laceration) or required the administration of blood products for patients with coagulopathy. "Death" indicated death resulted from the injuries from the fall and not the physiological events causing the fall.

For the analyses of general and fall-related characteristics, we included the falls of patients who fell only once and the first fall of patients who fell multiple times. Group comparisons (moderate or major injuries group versus minor or no injuries group) were performed using chi-squared and Fisher exact tests, as appropriate, for categorical outcomes and Mann-Whitney U test for continuous outcomes. The magnitude of the association between injury severity (moderate or major injuries group / minor or no injuries group) and activity during the fall was quantified using the odds ratio. Univariate logistic regression models were used to calculate the odds ratio and 95% confidence intervals (CIs). A p value $<0.05$ was considered statistically significant. All statistical analyses were performed with EZR, which is a modified version of R commander designed to add statistical functions frequently used in biostatistics [20]. This study was approved by the Research Ethics Committee of Asa Citizens Hospital (04-1-7) and the need for consent was waived by the ethics committee. This was a retrospective analysis of previously collected, non-identifiable information, and involved no change in the management of patients. Obtaining individual consent was not feasible so patients records were anonymized and de-identified prior to analysis. Data collected through medical records were managed such that no one other than four designed researchers could access them and anonymized data was used for analysis.

## Results

There were 318 patients with reported falls during the study period. Of those patients, 271 (85.2%) fell once during the study period and 47 (14.8%) fell more than once. Among the 318 patients, 268 (84.3%) had no related injury (none), 40 (12.6%) experienced minor injury, 3 (0.9%) experienced moderate injury, and 7 (2.2%) experienced major injury.

Table 1 presents a comparison of general and fall-related characteristics of all patients with reported falls. The correlation results among the two groups revealed no statistically significant difference in gender, assist type, shift when fall occurred, or fallen in the past 6 months. There was no significant difference in median (interquartile range) patient age or length of stay between the moderate or major injuries group and the minor or no injuries group. Compared with the minor or no injuries group, patients with moderate or major injuries group tended to experience their injuries in a place other than their bedroom (e.g., patient's bathroom, hallway, treatment room) and during ambulating. In the moderate or major injuries group, 4 falls occurred directly from a bed. In the minor or no injuries group, 129 falls occurred directly from a bed, 46 falls occurred directly from a bedside commode, 20 falls occurred directly from a chair, 19 falls occurred directly from a toilet, 15 falls occurred during standing (not trying

**Table 1. Comparison of general and fall-related characteristics between the moderate or major injuries group and minor or no injuries group.**

| Variable | Moderate or major injuries group (n = 10) | Minor or no injuries group (n = 308) | P-value |
|---|---|---|---|
| **Gender, n (%)** | | | 0.521 |
| Female | 6 (60.0) | 139 (45.1) | |
| Male | 4 (40.0) | 169 (54.9) | |
| **Age (year)** | 84.5 (79.75–90.75) | 81 (74–86) | 0.088 |
| **Length of Stay (day)** | 9.5 (8–13) | 15 (9.75–24) | 0.147 |
| **Assist type, n (%)** | | | >0.99 |
| Unassisted | 10 (100) | 303 (98.4) | |
| Assisted | 0 (0) | 5 (1.6) | |
| **Location of fall, n (%)** | | | 0.041* |
| Patient's bedroom | 5 (50.0) | 244 (79.2) | |
| Other | 5 (50.0) | 64 (20.8) | |
| **Shift when fall occurred, n (%)** | | | 0.055 |
| Day | 2 (20.0) | 162 (52.6) | |
| Evening / night | 8 (80.0) | 146 (47.4) | |
| **Fallen in past 6 months, n (%)** | | | 0.462 |
| YES | 1 (10.0) | 81 (26.3) | |
| NO | 9 (90.0) | 227 (73.7) | |
| **Activity during fall, n (%)** | | | 0.013* |
| Ambulating | 6 (60.0) | 69 (22.4) | |
| Other | 4 (40.4) | 239 (77.6) | |

Values are expressed as median(interquartile range) or number(%)

*Statistically significant differences

other action) and 10 falls occurred directly from a wheel chair. Univariate analysis showed that moderate or major injuries caused by a fall was associated with the activity during the fall (odds ratio: 5.20; CIs: 1.43–18.9, p = 0.013).

Of the 10 patients (3.1%) with 11 moderate or major injuries, 4 patients suffered lacerations requiring sutures, 5 patients had fractures (four vertebral fracture, one patella fracture) and 2 patients had brain injuries. Among the moderate or major injuries group, 6 of 10 patients (60.0%) were allowed to walk alone (Table 2).

## Discussion

The purpose of this study was to investigate the relationship between the moderate or major injuries caused by a fall and activity during the fall. The present study also found that falls occurring during ambulation caused moderate or major injuries in an acute care hospital. Because of risk of significant injury, increased costs, and emotional harms, reducing the number of falls in the hospital is a major priority [21]. Studies on falls in hospitals mainly investigated the characteristics of patients and falls [22]. Previous studies have reported that falls during walking are associated with fractures in a rehabilitation hospital [16]. Our study suggests that in an acute care hospital, falls during ambulating were associated not only with fractures, but also with lacerations that required sutures and resulted in brain injuries. Because of the influence of falling from a full height position or acceleration while walking, the severity of

Table 2. Characteristics of patients in the moderate or major injuries group.

| iD No. | Type of injury | Location of fall | Activity during fall | Walk alone |
|---|---|---|---|---|
| 1 | suturing, fracture | Patient's bedroom | ambulating | allowed |
| 2 | fracture | Patient's bedroom | reposioning in bed | allowed |
| 3 | suturing | Patient's bedroom | getting out of bed | not allowed |
| 4 | suturing | Hall way | ambulating | allowed |
| 5 | fracture | Patient's bathroom | ambulating | allowed |
| 6 | fracture | Patient's bathroom | dressing | allowed |
| 7 | fracture | Patient's bedroom | ambulating | not allowed |
| 8 | suturing | Patient's bathroom | ambulating | allowed |
| 9 | brain injury | Treatment room | getting out of bed | not allowed |
| 10 | brain injury | Patient's bedroom | ambulating | not allowed |

iD No. = Identification number of serious injury patient

the injury may be increased during ambulation. In this study, moderate or major injuries tended to occur in the evening/night, but without a significant difference. Generally, there are fewer nurses working during the evening/night than during the daytime, so the number of caregivers at evening/night is smaller. Kim et al. suggested that the incidence of serious injuries was higher in situations without a caregiver, but no significant difference was found [23]. This is consistent with our current findings. Among 10 patients of the moderate or major injuries, 6 were allowed to walk alone. This finding can offer guidance on future strategies to prevent fall related serious injury. Patients in an acute care hospital have an acute change in their body and mental functional status and environment. Francis-Coad reported that the injurious falls in an acute care hospital may reflect patients having significant functional decline early in admission when they are acutely ill and unfamiliar with their new environment [24]. In the study by Hitcho et al., many of the hospital patients were ambulating at the time of fall, usually unassisted [2]. Therefore, to prevent injuries caused by falls in an acute care hospital, it is important to evaluate patients' ambulating ability at several appropriate times depending on changes in body functional status during the patient's hospitalization. For example, the use of walking aids and environmental adjustments may be necessary and effective for patient safety. Furthermore, conducting a post-fall assessment in hospitalized older adults following a fall in order to identify the mechanism of the fall, any resulting injuries, any precipitating factors (such as new intercurrent illness, complications or delirium), to reassess the individual's fall risk factors, and adjust the intervention strategy accordingly is recommended [14].

The Joint Commission states that any patient of any age or physical ability can be at risk for a fall due to physiological changes resulting from a medical condition, medications, surgery, procedures, or diagnostic testing that can leave the patient weakened or confused [11]. To effectively prevent falls on wards, it is essential to assess and target all identified patient risk factors [25]. A history of falling is known as predictive factor of future falls in patients [26, 27]. Aryee et al. suggested that a fall history should be documented upon admission to the hospital and that fallers should be offered special precautions to prevent future injuries [28]. Our results showed that there was no significant difference in history of falls between the group of patients with moderate or major injuries and minor or no injuries group. Wellens et al., indicated that the history of falls may mask the influence of factors causing these earlier falls in a way that it can be considered as an indicator of an underlying problem like impaired balance [29]. Previous research has demonstrated that risk scores do not reliably predict which patients are at risk of falls or injurious falls [30, 31]. These results suggest that it is important to take adequate

safety measures for all patients to prevent fall related injury, even in low-risk inpatients. For example, randomized trials evaluating interventions that engage patients directly through education have shown reductions in both falls and injurious falls on subacute wards [32, 33].

About 60.0% of the patients in the moderate or major injuries group and 45.1% of the patients in the group with minor or no injuries were female. There was no significant difference between genders, which is consistent with the findings of a previous study [28]. In another prior study, age ≧80 years was a risk factor associated with hip fracture in a rehabilitation hospital [16]. In our study, fractures occurred in five patients and four of whom were older than 80 years. Despite fractures of the hip are a common fall related injury, no hip fractured occurred in this study [34]. Fractures may also be influenced by the timing and location, vertebral fracture can also occur spontaneously rather than falls-related [35]. Although increased age may be a risk for fall related fracture in the acute care hospital, we could not draw this conclusion because of the small sample size in our study, especially in the moderate or major injuries group. In some studies, no correlation was found between age or gender and patient falls [2, 36]. Future research needs to further examine the link between fall related injury and gender/age in acute care hospitals.

About 50.0% of the patients in the moderate or major injuries group and 20.8% of the patients in the group with minor or no injuries fell outside their bedroom. There was a significant difference between the moderate or major injuries group and the minor or no injuries group. In our study, falls that occurred outside the patient's bedroom were associated with an increased risk of fall related moderate or major injuries. Krauss et al. reported that falling in a patient care area other than the patient's bedroom was associated with an increased risk of fall-related injury [13].

The current study has some limitations. First, our study was performed at a single acute care hospital, and our findings therefore may not be generalized to other acute care hospitals. Second, the assessment of injury was based on the electronic medical records and interviews with the patients. Finally, the small sample size prevented us from estimating accurate odds ratios.

In conclusion, this study provides evidence that falls that occur during ambulation can cause moderate or major injuries in an acute care hospital. Our study suggests that falls during ambulating were associated not only with fractures, but also with lacerations that required sutures and resulted in brain injuries in the acute care hospital. Among the group with moderate or major injuries, more falls occurred outside the patient's bedroom than in the group with minor or no injuries. Therefore, based on our results, it is important to prevent fall-related moderate or major injuries while the patient is walking outside their bedroom. The injurious falls in an acute care hospital may reflect patients having significant functional decline early in admission when they are acutely ill and unfamiliar with their new environment [24]. Many of the hospital patients were ambulating at the time of fall, usually unassisted [2]. Therefore, to prevent injuries caused by falls in an acute care hospital, it is important to evaluate patients' ambulating ability at several appropriate times depending on changes in body functional status during the patient's hospitalization.

## Supporting information

**S1 Table. Group comparison for continuous outcomes.**
(XLSX)

**S2 Table. Group comparison for categorical outcomes.**
(XLSX)

**S3 Table. Univariate logistic regression models.**
(XLSX)

## Acknowledgments

We thank all of the patients in the study, and all of nurses who submitted reports on falls. The author would like to thank Dr. Keigo Dote who encouraged us to do this research.

## Author Contributions

**Conceptualization:** Kosuke Kobayashi.

**Data curation:** Kosuke Kobayashi.

**Formal analysis:** Kosuke Kobayashi.

**Funding acquisition:** Kosuke Kobayashi.

**Investigation:** Kosuke Kobayashi, Naohiro Kido, Shoji Wakabayashi, Kyoko Yamamoto.

**Methodology:** Kosuke Kobayashi.

**Project administration:** Kosuke Kobayashi, Masami Tamura.

**Resources:** Kosuke Kobayashi.

**Software:** Kosuke Kobayashi.

**Supervision:** Kosuke Kobayashi, Jun Hihara.

**Validation:** Kosuke Kobayashi.

**Visualization:** Kosuke Kobayashi, Tomoko Sakahara.

**Writing – original draft:** Kosuke Kobayashi.

**Writing – review & editing:** Kosuke Kobayashi.

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
