## [Decision Letter · Decision Letter 0]

3 May 2023

PONE-D-22-34327Association between fall-related serious injury and activity during fall in an acute care hospitalPLOS ONE

Dear Dr. Kobayashi,

Thank you for submitting your manuscript to PLOS ONE. After careful consideration, we feel that it has merit but does not fully meet PLOS ONE’s publication criteria as it currently stands. Therefore, we invite you to submit a revised version of the manuscript that addresses the points raised during the review process.

We apologize for the delay in the review process as it was difficult to select appropriate reviewers. Your manuscript has been peer reviewed by two reviewers. As a result, several significant issues have been identified. Please address all review comments point by point by appropriately revising or responding to them. When revising the manuscript, please clearly state in your response why you revised or not in that way. Please highlight the revisions made in the manuscript. These efforts would greatly help in the second review process.

We look forward to receiving your revised manuscript.

Kind regards,

Masaki Tago, M.D., Ph.D., FACP.

Academic Editor

PLOS ONE

Journal Requirements

4. Please include your tables as part of your main manuscript and remove the individual files. Please note that supplementary tables (should remain/ be uploaded) as separate "supporting information" files

Additional Editor Comments:

Thank you for submitting your manuscript to PLOS ONE. We apologize for the delay in the review process as it was difficult to select appropriate reviewers. Your manuscript has been peer reviewed by two reviewers. As a result, several significant issues have been identified. Please address all review comments point by point by appropriately revising or responding to them. When revising the manuscript, please clearly state in your response why you revised or not in that way. Please highlight the revisions made in the manuscript. These efforts would greatly help in the second review process.

Reviewers' comments:

Reviewer's Responses to Questions

**Comments to the Author**

1. Is the manuscript technically sound, and do the data support the conclusions?

Reviewer #1: Yes

Reviewer #2: Partly

2. Has the statistical analysis been performed appropriately and rigorously? 

Reviewer #1: Yes

Reviewer #2: Yes

3. Have the authors made all data underlying the findings in their manuscript fully available?

Reviewer #1: Yes

Reviewer #2: Yes

4. Is the manuscript presented in an intelligible fashion and written in standard English?

Reviewer #1: Yes

Reviewer #2: Yes

5. Review Comments to the Author

Reviewer #1: A useful analysis of differences between injurious and non/low-injurious falls and fallers in a single acute hospital.

References used in introduction and discussion include fairly small and old studies rather than wider and more recent sources e.g. Cochrane systematic review and world guidelines https://www.bgs.org.uk/wfg which would be better sources.

Second falls are excluded to 'reduce bias' but it is unclear what that bias would be. It would be more typical to analyse some data by fallers including multiple and single fallers (e.g. age and gender) and some factors that will vary even for the same faller by falls (e.g. injury from fall, and activity before falling). Whilst not suggesting the study should be redone, the authors do need to describe more clearly why they excluded all but 'first falls'.

Whilst the study clearly explains that 'serious' for the purpose of the study included moderate as well as major injury the term 'serious' implies significant injury, and using a term that is more intuitive or simply using an un-shortened 'moderate or major injury' versus 'minor or no injury' throughout would aid the reader's understanding.

Given falls typically are unwitnessed and often affect patients with cognitive deficits, methods need to explain how the patients' activity at time of fall was known.

The majority of non-injurious or minor injury falls are recorded to have happened other than when ambulating - 77% 'other' is a surprisingly high percentage and would be helpful to at least summarise what 'other' typically were (falls directly from a bed or chair?)

Fracture types described are atypical for falls in older people in hospital where Colles' fracture, humerus, pubic rami and hip fracture more often encountered - likely to be chance but does merit discussion given vertebral fracture can be spontaneous rather than falls-related.

To an international audience very surprising to find almost all falls (99%) were in patients who did not need assistance walking and needs some explanation as to how apparently so very few patients unsafe to walk alone ever did so.

Given the significance differences found in falls in room/outside room helpful to have context of whether rooms in the hospital typically ensuite or not. I think from table 2 'patient's room' might mean the bedroom only, and if so perhaps clearer to use the term 'patient's bedroom'.

Patient involvement in the design of the study and the write-up should be described.

Reviewer #2: In the discussion section, there are logical leaps between the interpretation of the results and the causes of the results.

Major:

1. Page 8, row 180-181: The authors described that "serious injury resulting from a fall occurred when the patient was in an unassisted situation". However, I didn’t find a significant difference between them in Table 1. Therefore, based on the current study results, the sentence was inappropriate.

2. Page 8, row 192-193: Please provide any evidence to support the sentence "Patients in an acute care hospital have an acute change in their body and mental functional status and environment", the logic that the change causes falls while walking, and the sentence "Therefore, to prevent injuries caused by falls, it is important to evaluate patients’ ability to walk at several appropriate times".

3. Page 9, row 197-204: This paragraph mentioned the possibility for difference between fall rates of inpatients with injuries in previous studies and the current study. However, the rates were similar with each other and the rates were not statistically judged, and the subjects of those studies were different, which were not considered in the logic. The discussion in this paragraph lacked evidence.

4. Page 9, row 211-213: The authors said that "Our results showed that the group of patients with serious injuries were less likely to have a history of falls". However, the p-value was 0.462 in Table 1, which was not low even relatively. It could not be said that they were "less likely".

5. Page 10, row 238-239: The sentence "As previously reported in a study of injured athletes, video analysis can provide insights into the mechanisms of injury and can help to devise preventative strategies [34]." seemed not to be relevant to the current study results. I suggest to delete this sentence.

6. Page 11, row 249-251: The statement "Because the patient's condition is easily changed in an acute care hospital, it is important to evaluate the patients’ ability to walk at several appropriate times to prevent injury caused by falls." had the problem as the same as ②, which had a logical leap. Please revise appropriately.

6. PLOS authors have the option to publish the peer review history of their article (what does this mean?). If published, this will include your full peer review and any attached files.

Reviewer #1: **Yes: **Dr Frances Healey

Reviewer #2: No

---

## [Author Response · Author response to Decision Letter 0]

26 May 2023

Thank you very much for providing important comments. We are thankful for the time and energy you expended. We are delighted to hear that you think our work will spark debate in our field. In the following sections, you will find our responses to each of your points and suggestions. We are grateful for the time and energy you expended on our behalf.

Dear Mr. Masaaki Tago

Thank you for your telling us about the PLOS ONE’s style requirements file. We checked both pdf files and revised. 

Thank you for pointing it out. We have reflected this comment as Page 6, row 138-144.

Thank you for your telling us about the ORCID. Corresponding author create a new iD (0009-0001-2070-8888). 

4. Please include your tables as part of your main manuscript and remove the individual files. Please note that supplementary tables (should remain/ be uploaded) as separate "supporting information" files

Thank you for pointing it out. We have included two tables as part of main manuscript.

Dear Dr. Frances Healey

1. References used in introduction and discussion include fairly small and old studies rather than wider and more recent sources e.g. Cochrane systematic review and world guidelines https://www.bgs.org.uk/wfg which would be better sources.

Thank you for your suggestion. We have used both Cochrane systematic review and world guidelines as references. (Page 4, row 83-85, Page 10, row 210-214)

2. Second falls are excluded to 'reduce bias' but it is unclear what that bias would be. It would be more typical to analyse some data by fallers including multiple and single fallers (e.g. age and gender) and some factors that will vary even for the same faller by falls (e.g. injury from fall, and activity before falling). Whilst not suggesting the study should be redone, the authors do need to describe more clearly why they excluded all but 'first falls'.

Thank you for providing these insights. We agree with you and have incorporated this suggestion. (Page 5, row 105-109)

3. Whilst the study clearly explains that 'serious' for the purpose of the study included moderate as well as major injury the term 'serious' implies significant injury, and using a term that is more intuitive or simply using an un-shortened 'moderate or major injury' versus 'minor or no injury' throughout would aid the reader's understanding.

Thank you for your suggestion. We agree with you and have incorporated this suggestion throughout our paper.

4. Given falls typically are unwitnessed and often affect patients with cognitive deficits, methods need to explain how the patients' activity at time of fall was known.

We agree with you. We have reflected this comment as Page 5, row 115-117.

5. The majority of non-injurious or minor injury falls are recorded to have happened other than when ambulating - 77% 'other' is a surprisingly high percentage and would be helpful to at least summaries what 'other' typically were (falls directly from a bed or chair?)

This is an interesting perspective. We have reflected this comment as Page 7, row 159-163.

6. Fracture types described are atypical for falls in older people in hospital where Colles' fracture, humerus, pubic rami and hip fracture more often encountered - likely to be chance but does merit discussion given vertebral fracture can be spontaneous rather than falls-related.

We agree with your assessment. We have reflected this comment as Page 11, row 236-238.

7. To an international audience very surprising to find almost all falls (99%) were in patients who did not need assistance walking and needs some explanation as to how apparently so very few patients unsafe to walk alone ever did so.

This is an interesting perspective. In this study, “unassisted” situation means that there was no one when the fall occurred. Many patients who fell in this study told to use the call light before they move. We hope that you agree.

8. Given the significance differences found in falls in room/outside room helpful to have context of whether rooms in the hospital typically ensuite or not. I think from table 2 'patient's room' might mean the bedroom only, and if so perhaps clearer to use the term 'patient's bedroom'.

Thank you for your suggestion. We agree with you and have incorporated this suggestion throughout our paper.

9. Patient involvement in the design of the study and the write-up should be described.

Thank you for providing these insights. We have reflected this suggestion as Page 5, row99-101, 105-109, 116-117, 139-142. We hope that you agree.

Dear Reviewer #2

1. Page 8, row 180-181: The authors described that "serious injury resulting from a fall occurred when the patient was in an unassisted situation". However, I didn’t find a significant difference between them in Table 1. Therefore, based on the current study results, the sentence was inappropriate.

We agree with your assessment. We have totally deleted the sentence.

2. Page 8, row 192-193: Please provide any evidence to support the sentence "Patients in an acute care hospital have an acute change in their body and mental functional status and environment", the logic that the change causes falls while walking, and the sentence "Therefore, to prevent injuries caused by falls, it is important to evaluate patients’ ability to walk at several appropriate times".

Thank you for your suggestion. We have reflected this suggestion as Page 10, row203-209. We hope that you agree.

3. Page 9, row 197-204: This paragraph mentioned the possibility for difference between fall rates of inpatients with injuries in previous studies and the current study. However, the rates were similar with each other and the rates were not statistically judged, and the subjects of those studies were different, which were not considered in the logic. The discussion in this paragraph lacked evidence.

We agree with your assessment. We have totally deleted the sentence.

4. Page 9, row 211-213: The authors said that "Our results showed that the group of patients with serious injuries were less likely to have a history of falls". However, the p-value was 0.462 in Table 1, which was not low even relatively. It could not be said that they were "less likely".

We agree with your suggestion. We have reflected this suggestion as Page 10-11, row221-225. We hope that you agree.

5. Page 10, row 238-239: The sentence "As previously reported in a study of injured athletes, video analysis can provide insights into the mechanisms of injury and can help to devise preventative strategies [34]." seemed not to be relevant to the current study results. I suggest to delete this sentence.

We have totally deleted the sentence.

6. Page 11, row 249-251: The statement "Because the patient's condition is easily changed in an acute care hospital, it is important to evaluate the patients’ ability to walk at several appropriate times to prevent injury caused by falls." had the problem as the same as ②, which had a logical leap. Please revise appropriately.

We have reflected this suggestion as Page 12, row261-267. We hope that you agree.

Again, we appreciate all of your insightful comments. We worked hard to be responsive to them. Thank you for taking the time and energy to help us improve the paper.

Best regards,

Kosuke Kobayashi, Ph.D, PT

---

## [Decision Letter · Decision Letter 1]

26 Jun 2023

Association between fall-related serious injury and activity during fall in an acute care hospital

PONE-D-22-34327R1

Dear Dr. Kobayashi,

We’re pleased to inform you that your manuscript has been judged scientifically suitable for publication and will be formally accepted for publication once it meets all outstanding technical requirements.

Kind regards,

Masaki Tago, M.D., Ph.D., FACP.

Academic Editor

PLOS ONE

Additional Editor Comments (optional):

Reviewers' comments:

Reviewer's Responses to Questions

**Comments to the Author**

1. If the authors have adequately addressed your comments raised in a previous round of review and you feel that this manuscript is now acceptable for publication, you may indicate that here to bypass the “Comments to the Author” section, enter your conflict of interest statement in the “Confidential to Editor” section, and submit your "Accept" recommendation.

Reviewer #1: All comments have been addressed

Reviewer #2: All comments have been addressed

2. Is the manuscript technically sound, and do the data support the conclusions?

Reviewer #1: Yes

Reviewer #2: Yes

3. Has the statistical analysis been performed appropriately and rigorously? 

Reviewer #1: Yes

Reviewer #2: Yes

4. Have the authors made all data underlying the findings in their manuscript fully available?

Reviewer #1: Yes

Reviewer #2: Yes

5. Is the manuscript presented in an intelligible fashion and written in standard English?

Reviewer #1: Yes

Reviewer #2: Yes

6. Review Comments to the Author

Reviewer #1: Happy that past comments have been addressed. Happy that past comments have been addressed. (note 100 characters were required so repeated this sentence)

Reviewer #2: (No Response)

7. PLOS authors have the option to publish the peer review history of their article (what does this mean?). If published, this will include your full peer review and any attached files.

Reviewer #1: **Yes: **Dr Frances Healey

Reviewer #2: No

---

## [Editor Report · Acceptance letter]

29 Jun 2023

PONE-D-22-34327R1 

Association between fall-related serious injury and activity during fall in an acute care hospital 

Dear Dr. Kobayashi:

I'm pleased to inform you that your manuscript has been deemed suitable for publication in PLOS ONE. Congratulations! Your manuscript is now with our production department. 

Kind regards, 

on behalf of

Dr. Masaki Tago 

Academic Editor

PLOS ONE